# Co-Existence of *bla*_NDM-1_, *bla*_OXA-23_, *bla*_OXA-64_, *bla*_PER-7_ and *bla*_ADC-57_ in a Clinical Isolate of *Acinetobacter baumannii* from Alexandria, Egypt

**DOI:** 10.3390/ijms241512515

**Published:** 2023-08-07

**Authors:** Sandra Sánchez-Urtaza, Alain Ocampo-Sosa, Ainhoa Molins-Bengoetxea, Jorge Rodríguez-Grande, Mohammed A. El-Kholy, Marta Hernandez, David Abad, Sherine M. Shawky, Itziar Alkorta, Lucia Gallego

**Affiliations:** 1Department of Immunology, Microbiology and Parasitology, Faculty of Medicine and Nursing, University of the Basque Country, 48940 Leioa, Spain; sandrasanchezurtaza@gmail.com (S.S.-U.); ainhoamolinsb@gmail.com (A.M.-B.); 2Microbiology Service, Health Research Institute (IDIVAL), University Hospital Marqués de Valdecilla, 39008 Santander, Spain; jorgergrande@gmail.com; 3CIBER de Enfermedades Infecciosas (CIBERINFEC), Instituto de Salud Carlos III, 28029 Madrid, Spain; 4Division of Clinical and Biological Sciences, Department of Microbiology and Biotechnology, College of Pharmacy, Arab Academy for Science, Technology & Maritime Transport (AASTMT), Alexandria P.O. Box 1029, Egypt; mohammed.elkholy@aast.edu; 5Laboratory of Molecular Biology and Microbiology, One Health, Agrarian Technological Institute of Castile and Leon (ITACyL), 47009 Valladolid, Spain; hernandez.marta@gmail.com (M.H.); dabad87@gmail.com (D.A.); 6Medical Research Institute, Alexandria University, Alexandria 5422031, Egypt; sherineshawky@hotmail.com; 7Department of Biochemistry and Molecular Biology, Faculty of Science and Technology, University of the Basque Country, 48940 Leioa, Spain; itzi.alkorta@ehu.eus

**Keywords:** *Acinetobacter baumannii*, carbapenemase genes, whole genome sequencing

## Abstract

The increasing rates of antimicrobial resistance among carbapenem-resistant *Acinetobacter baumannii* in the Middle East and North Africa are one of the major concerns for healthcare settings. We characterised the first *A. baumannii* isolate harbouring five β-lactamases identified in Egypt. The isolate Ale25 was obtained from an ICU patient of a hospital from Alexandria. The isolate was phenotypically and genotypically screened for carbapenemase genes. The isolate was resistant to carbapenems, aminoglycosides, fluoroquinolones and cefiderocol. Whole-Genome Sequencing identified five β-lactamase genes, *bla*_NDM-1_, *bla*_OXA-23_, *bla*_OXA-64_, *bla*_PER-7_ and *bla*_ADC-57_, together with other antibiotic resistance genes, conferring resistance to sulfonamides, macrolides, tetracyclines, rifamycin and chloramphenicol. Virulome analysis showed the presence of genes involved in adhesion and biofilm production, type II and VI secretion systems, exotoxins, etc. Multi-Locus Sequence Typing analysis identified the isolate as Sequence Types 113^Pas^ and 2246^Oxf^, belonging to International Clone 7. Sequencing experiments revealed the presence of four plasmids of 2.7, 22.3, 70.4 and 240.8 Kb. All the β-lactamase genes were located in the chromosome, except the *bla*_PER-7_, gene which was found within the plasmid of 240.8 Kb. This study highlights the threat of the emergence and dissemination of these types of isolates.

## 1. Introduction

Carbapenem-resistant *Acinetobacter baumannii* (CRAB) constitutes one of the major challenges for healthcare settings due to its high rates of antimicrobial resistance, heading the list of critical pathogens published by The World Health Organization. The main mechanisms of carbapenem resistance in *A. baumannii* are OXA-type carbapenemases [1]. However, the number of *A. baumanni* isolates harbouring class B metallo-β-lactamases, such as New Delhi Metallo-Beta-lactamases (NDM), have dramatically increased [1]. In the Middle East and North Africa, *A. baumannii* clinical outbreaks are usually caused by isolates endemically producing carbapenemases NDM and OXA-23 [2]. Furthermore, in *A. baumannii* clinical isolates, class A β-lactamases, such as PER-type β-lactamases, can also be responsible for resistance to carbapenems and the last-resort antibiotic cefiderocol [3]. Furthermore, as demonstrated by other authors, a plasmid containing *bla*_PER-1_ gene was transferred to a susceptible *A. baumannii* CIP70.10 resulted in a ≥16-fold increase in the MIC of cefiderocol [3,4]. Despite the alarming situation, there is little information or studies about *A. baumannii* in Egypt [2]. The aim of the present study was the molecular characterisation of the first *A. baumannii* isolate harbouring five β-lactamases in Egypt, obtained from a 63-year-old renal dialysis female patient who entered the ICU of a hospital from Alexandria.

## 2. Results

The isolate was identified by MALDI-TOF, *gyrB* multiplex PCR and was confirmed by whole genome sequencing as *A. baumannii,* and it also tested positive for *bla*_OXA-51-like_ gene.

The susceptibility testing results are shown in Table 1.

Molecular typing assigned the isolate to STs 113 by Pasteur Scheme and ST 2246 by Oxford Scheme. These results in combination with the *bla*_OXA-51-like_ variant revealed that the isolate belonged to IC7.

Three recognised carbapenemase genes were detected by Whole Genome Sequencing, *bla*_OXA-64_, *bla*_NDM-1_ and *bla*_OXA-23_, as well as the Extended Spectrum Beta-Lactamase coding gene *bla*_PER-7_ and the cephalosporinase gene *bla*_ADC-57_. The isolate also harboured several aminoglycoside resistance genes conferring resistance to streptomycin, gentamycin, tobramycin, amikacin, kanamycin and spectinomycin, as well as genes conferring resistance to sulfonamides, macrolides, tetracyclines, rifamycin and chloramphenicol, respectively (Table 2).

Analysis of the genetic surroundings of the β-lactamase genes located the *bla*_NDM-1_ within the truncated isoform of transposon Tn*125* (ΔTn*125*) (Figure 1a). The *bla*_PER-7_ gene was found within a complex structure connecting the IS*CR1* element and a class 1 integron with part of IS*26* upstream of the integron. The gene-cassette variable regions contained the antibiotic resistance genes *arr-2* and *cmlA5* genes (Figure 1b). Upstream of *bla*_PER-7_, we found the IS*CR1* element. A *gst* gene and an *abc* transporter gene were detected in the IS*CR1* linked genes variable region downstream of *bla*_PER-7_. Interestingly, another IS*CR1* element, IS*5,* and part of IS*10A* were found downstream of the 3′-CS. The *bla*_OXA-23_ gene was located within transposon Tn*2006* (Figure 1c). No insertion sequences were found in close proximity to *bla*_OXA-64_ or *bla*_ADC-57_ genes (Figure 1d,e).

Virulome analysis revealed the presence of a wide variety of genes involved in adherence and biofilm production (type IV pili, biofilm-controlling response regulator, AdeFGH efflux pump, quorum sensing, Csu fimbriae, biofilm-associated protein, outer-membrane protein, Poly-N-acetyl-D-glucosamine), immune modulation (capsule, lipopolysaccharide and penicillin-binding Protein G), effector delivery systems (type II and type VI secretion systems), exotoxins (phospholipases C and D) and iron uptake (acinetobactin) (Table 3).

Two structures of around 26 and 2.4 Kb (Figure 2A) and a plasmid of approximately 240 Kb (Figure 2B) were observed by conventional plasmid extraction and S1-Pulsed Field Gel Electrophoresis experiments, respectively. Further sequencing experiments revealed three conjugative plasmids of 22.3, 70.4 and 240.8 Kb as well as a non-conjugative plasmid of 2.7 Kb. PCR-based and whole-genome-sequence-based *Acinetobacter* Replicon typing experiments identified *repAci6* (homology group GR6) in the 70.4 Kb plasmid and a *rep* gene encoding a Rep_3 family protein R3-T60 in the 240.8 Kb plasmid. We were not able to identify the *rep* genes of the 2.7 and 22.3 Kb plasmids using these methods.

Plasmid sequencing experiments located all the β-lactamase genes in the chromosome (Appendix A Appendix A), except the *bla*_PER-7_ gene, which was found within the 240.8 Kb plasmid pAbAle25.1 together with aminoglycoside (*armA, strA, strB*), sulfonamide (*sul1, sul2*), macrolide (*msrE, mphE*), tetracycline (*tet(B)*), chloramphenicol (*cmlA5*) and rifamycin (*arr-2*) resistance genes (Figure 3). The aminoglycoside resistance gene *aph (3′)-VI* was found within the 70.4 Kb plasmid. Hybridisation experiments also confirmed the chromosomal location of the *bla*_NDM-1_ gene (Figure 2B).

## 3. Discussion

Consistent with the recent studies reporting a carbapenem resistance rate of 98% among Egyptian *A. baumannii* isolates [5], our isolate was resistant to imipenem and meropenem. To our knowledge, no cefiderocol-resistant *A. baumannii* isolates have been previously reported in Egypt, turning the spotlight on the emergence and dissemination of resistance to cefiderocol in Egypt.

The isolate belonged to IC7, which is especially interesting considering that isolates belonging to IC7 are frequently reported in South America [6,7] but not in Egypt, where IC2 is the most prevalent international clone [8].

The presence of IS*Aba125* upstream of the *bla*_NDM-1_ may enhance its expression [9]. In our isolate, *bla*_NDM-1_ gene was located in the chromosome, which is the most frequent localization of these genes, as previously reported by other authors [1,10]. By BLASTn, we observed that the genetic environment of the *bla*_NDM-1_ gene was 100% similar to an *A. baumannii* isolated in France in 2011 (Acc. No. JX000237.3), an *A. baumannii* isolated in Lebanon in 2015 (Acc. No. CP082952.1) and an *A. baumannii* isolated in India in 2018 (Acc. No. CP038644) assigned to ST85 Pasteur Scheme, ST1089 Oxford Scheme with *bla*_OXA-94_, and which harboured a single copy of *bla*_NDM-1_. These three isolates belonged to IC9, suggesting that this genetic context is circulating and being transmitted among different ICs.

The structure containing the IS*CR1* element and a class 1 integron, in which the *bla*_PER-7_ gene was located, has been previously described, and it is closely related to multidrug-resistant bacteria [11]. The presence of this betalactamase gene in a high-molecular-weight conjugative plasmid of 240.8 Kb has not been previously reported, although the *bla*_PER-1_ variant has been described by other authors within a similar structure [12]. Our concern is that it is the first time one obtains the sequence of this structure coding for multiple resistance and virulence genes. Despite the fact that the carbapenemase activity of PER-type and ADC-like enzymes is doubtful, recent studies have shown that PER-7 may exhibit carbapenemase activity [9] and that ADC-57 may hydrolyse ertapenem [13], which, in our isolate, may be contributing to carbapenem resistance.

The *bla*_OXA-23_ gene was found within the transposon Tn2006, which is the most frequently reported transposon harbouring this gene and the only one that has been experimentally proven to transpose [14,15]. The presence of IS*Aba1* upstream of the *bla*_OXA-23_ gene might lead to the overexpression and mobilisation of this carbapenemase gene, enhancing carbapenem resistance [16].

The results we obtained by different plasmid extraction methods showed that in order to obtain the complete profile, a combination of all of them is needed, as this improves the number of structures that can be detected. Each technique is not able to provide a complete landscape of the plasmidome that was finally revealed by sequencing experiments. Plasmid extraction kits and conventional lysis are useful for low-molecular-weight plasmids isolation and for a first screening, but it becomes necessary to complete these results with additional experiments, such as S1-PFGE or sequencing techniques. The combination of using short and long-read technologies to perform hybrid assemblies has been demonstrated to be a powerful tool for plasmids resolution [17].

To the best of our knowledge, only a single isolate harbouring four different classes of β-lactamase genes has been previously reported in Bangladesh [18], so this would be the first report of an *A. baumannii* clinical isolate harbouring four different classes of β-lactamase genes in Egypt, and it would also be the first cefiderocol-resistant isolate reported in Egypt. This study highlights the ability of *A. baumannii* to acquire and accumulate multiple antibiotic resistance genes, and it also turns the spotlight on the emergence and threat of the dissemination of IC7 isolates, not frequently reported in Egypt, and the importance of controlling the dissemination of these types of isolates.

## 4. Materials and Methods

A 63-year-old renal dialysis female patient with acute-on-chronic kidney disease was admitted to the hospital and entered the ICU on 13 November 2020. On 29 November 2020, a swab from around the abdominal pigtail catheter was obtained by the Microbiology Service, Alexandria University Medical Research Institute. Being a renal patient, she was treated with tigecycline injection as well as topical polymyxin. The patient was discharged home after 40 days at the ICU.

Species identification was assessed by matrix-assisted laser desorption/ionisation-time of flight mass spectrometry (MALDI-TOF/Vitek-MS with SARAMIS MS-IVD v2, Biomérieux, Marcy-l’Étoile, France), and the detection of *bla*_OXA-51-like_ gene was conducted by PCR [19,20,21] and *gyr*B multiplex PCR [22] and confirmed by whole genome sequencing. Minimal inhibitory concentrations of ticarcillin, ticarcillin/clavulanic acid, piperacillin, piperacillin/tazobactam, imipenem, meropenem, gentamicin, tobramycin, ciprofloxacin, aztreonam, tigecycline, cefiderocol, minocycline, colistin and trimethoprim/sulfamethoxazole were determined by broth microdilution method following CLSI guidelines. MICs were interpreted using the resistance breakpoints for *Acinetobacter* spp. from EUCAST (Version 13.1, June 2023, http://www.eucast.org/clinical_breakpoints/, accessed on 14 July 2023). Antimicrobial activity of cefiderocol was also determined by disk diffusion method using 30 µg cefiderocol discs (ThermoFisher Scientific, Waltham, MA, USA). *Escherichia coli* ATCC 25922 and *Pseudomonas aeruginosa* ATCC 27853 were used as control strains.

In order to know the genetic basis of antimicrobial resistance, carbapenemase-encoding genes were first analysed by three multiplex PCRs, including primers for *bla*_OXA-23-like, −40-like, −51-like, −58-like, −143-like_, _−235-like_, *bla*_VIM_, *bla*_KPC_, *bla*_NDM_, *bla*_OXA-48_, *bla*_IMI_, *bla*_GES_, *bla*_GIM_, *bla*_IMP_ and ISAba-1/*bla*_OXA-51-like_ [21,23]. Then, the Ale25 genome was fully sequenced using both the Illumina Miseq system with the v3 chemistry for 2 × 300 paired-end libraries (Illumina Inc., San Diego, CA, USA) [24] and the MinION Mk1C sequencing device with the Rapid Barcoding Kit (SQK-RBK004) with an R9 flow cell (FLO-MIN106) (Oxford Nanopore Technologies, Cambridge, UK).

Total DNA was purified with the DNeasy Blood and Tissue Kit (Qiagen, Hilden, Germany) and sequenced on a Illumina MiSeq device using reagents kit v3 for 2 × 300 paired-end libraries (Illumina Inc., San Diego, CA, USA) [24]. Raw reads from the sequencing platform were directly analysed using the in-house bioinformatics pipeline TORMES^®^ [25]. *A. baumannii* ATCC 17978 was used as a reference strain. Quality control and filtering of the reads were assessed using Trimmomatic [26], Prinseq [27] and Kraken [28]. Genome assembly was performed with SPAdes [29] and Quast [30] and genome annotation with Prokka software tool version 1.14.6 [31]. Multi-Locus Sequence Typing (MLST) was performed following Pasteur and Oxford typing schemes using Ridom SeqSphere+ software version 8.5.1 (Ridom© GmbH, Münster, Germany). The *bla*_OXA-51-like_ variant combined with the Sequence Type (ST) were used to assign the isolate to an International Clone (IC), as these variants have been previously described as being related to ICs [32]. The search for antibiotic resistance genes was conducted using Beta-Lactamase DataBase (BLDB) [33], NCBI BLAST [34] and ABRicate (Seemann T, Abricate, Github https://github.com/tseemann/abricate, accessed on 14 July 2023) against ResFinder [35], CARD [36] and ARG-ANNOT [37] databases. Genetic environments of the β-lactamase genes were edited and visualised by the use of SnapGene Viewer 6.0.5. Virulence factors were screened using Virulence Factors Database (VFDB) search tool [38] and Ridom SeqSphere+ software (Ridom© GmbH).

Basecalling and barcoding of Oxford Nanopore Technology reads was performed using guppy v.6.5.7+ca6d6af and minimap2 version 2.24-r112. Basecalling was conducted using super-quality mode. Quality checks and filtering of reads shorter than 250 pb or with an average quality under Q10 was carried out with NanoPack v.1.4.1 [39]. Read contamination was also checked by using Kraken v.2.1.2 8 [40]. The reads were assembled by using Flye v2.9.2-b1786 [41] and polished using medaka v.1.7.2 (https://github.com/nanoporetech/medaka, accessed on 14 July 2023). Assembly completion was inspected visually using Bandage [42], and quality was checked using Quast v.5.2.0 [30]. Then, assemblies were annotated using Bakta v.1.7.0 [43]. Abricate v1.0.1 (https://github.com/tseemann/abricate, accessed on 14 July 2023) was used to find antimicrobial resistance genes by using the following databases: NCBI-AMRFinderPlus [44], CARD [45], ResFinder [35] and ARG-ANNOT [37]. Finally, plasmids were detected and classified by using MOBsuite:3.0.3 [46] and Copla.py v.1.0 [47]. Plasmids were plotted by using Proksee.ca [48]. Genome sequences of Ale25 were generated by combining data from both the Illumina and MinION datasets using Unicycler v0.5.0 (https://github.com/rrwick/Unicycler, accessed on 14 July 2023). Finally, both *A. baumannii* PCR-based replicon typing (AB-PBRT) and *Acinetobacter* Plasmid Typing database were used against the draft genome to determine the presence of replicase genes, as previously described [49,50]. This assembled genome was submitted to NCBI under the accession number JANBZS000000000.

To investigate the presence of plasmids, we first performed plasmid extractions using GeneJET Plasmid Miniprep Kit (ThermoFisher Scientific, Waltham, MA, USA), following the manufacturer’s indications, and S1-Pulsed-Field Gel Electrophoresis (PFGE) experiments. The Bacterial DNA embedded in agarose plugs was digested using 14 units of S1-nuclease (Takara Bio, Kusatsu, Japan) per plug, and then it was run on a CHEF-DR III system (Bio-Rad, Munich, Germany) for 18 h at 6 V/cm and 14 °C. CHEF DNA Size Standard Lambda Ladder (Bio-Rad) was used as a molecular weight marker. Southern-blot hybridisation of the resulting PFGE was performed to locate *bla*_NDM-1_ gene with specific digoxigenin-labelled DNA probes (Roche, Mannheim, Germany). Signal detection was performed using DIG Nucleic Acid Detection Kit (Roche).

## Figures and Tables

**Figure 1 ijms-24-12515-f001:**
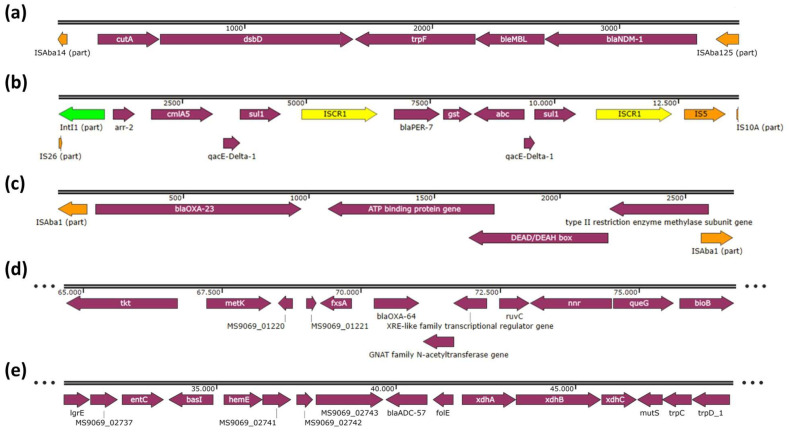
Genetic environments of the β-lactamase genes harboured by the isolate Ale25: (**a**) *bla*_NDM-1_; (**b**) *bla*_PER-7_; (**c**) *bla*_OXA-23_; (**d**) *bla*_OXA-64_; (**e**) *bla*_ADC-57_. The numbers above the arrows represent the base pairs of the sequences. The three dots in (**d**,**e**) represent that the sequences are larger than the fragment showed.

**Figure 2 ijms-24-12515-f002:**
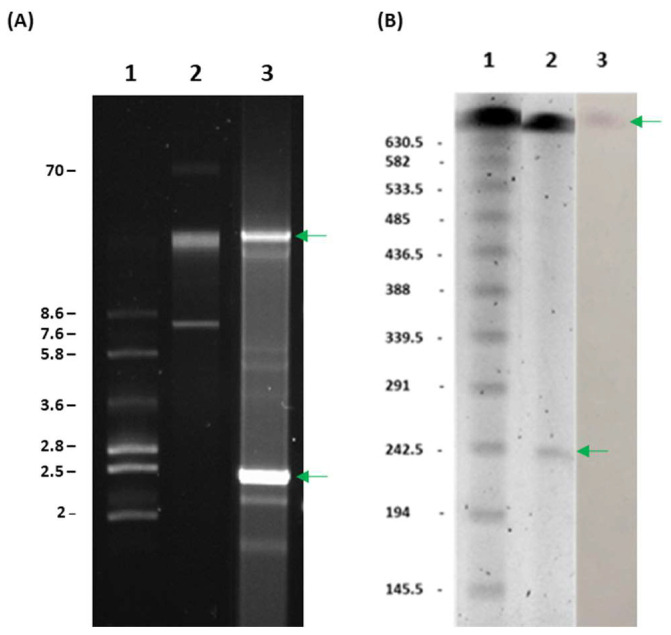
(**A**) Conventional plasmid extraction of the isolate Ale25. Lanes 1 & 2: control strains *E. coli* 678 (NCTC 50193) and 679 (NCTC 50192); lane 3: Ale25 profile showing plasmids of ~26 Kb and 2.4 Kb (green arrows); (**B**) S1 nuclease—Pulsed Field Gel Electrophoresis of the isolate Ale25. Lane 1: molecular weight marker; Lane 2: chromosome and a plasmid of ~240 Kb (green arrow); Lane 3: southern-blot hybridisation with a *bla*_NDM-1_ DNA probe showing the chromosomal location of the gene (green arrow). Molecular weights are expressed in Kb.

**Figure 3 ijms-24-12515-f003:**
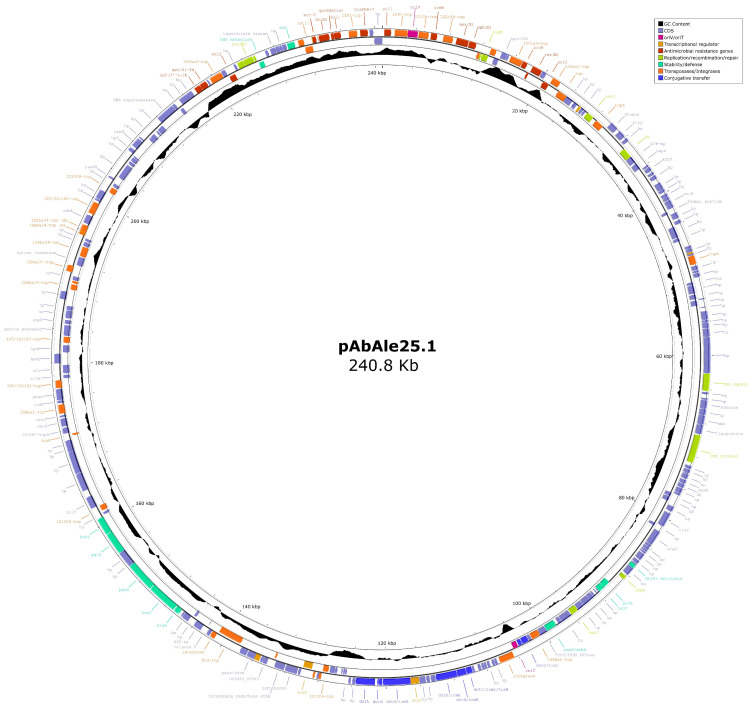
Plasmid pAbAle25.1 of 240.8 Kb harbouring, among other virulence genes, *bla*_PER-7_ gene.

**Table 1 ijms-24-12515-t001:** Minimum Inhibitory Concentrations (MICs) and interpretations of the *A. baumannii* isolate Ale25. “R” and “S” corresponds to resistant and susceptible, respectively. “IE” indicates that there is insufficient evidence that the organism or group is a good target for therapy with the agent.

Antibiotic	MIC (mg/L)	Interpretation
Ticarcillin/clavulanic acid	>64	IE
Piperacillin	>64	IE
Piperacillin/tazobactam	>64	IE
Imipenem	>64	R
Meropenem	>64	R
Amikacin	>64	R
Gentamicin	>64	R
Tobramycin	>64	R
Ciprofloxacin	32	R
Trimethoprim/sulfamethoxazole	>32	R
Tigecycline	0.125	IE
Minocycline	0.5	IE
Colistin	0.125	S
Cefiderocol	16	R

**Table 2 ijms-24-12515-t002:** Resistome of the *A. baumannii* isolate Ale25. Genes are represented in italics.

Β-Lactamase Genes	Tetracyclines	Sulfonamides	Aminoglycosides	Macrolides	Rifamycin	Chloramphenicol
*bla* _OXA-64_ ^1^	*tet(B)*	*sul1*	*armA*	*mph(E)*	*arr-2*	*cmlA5*
*bla* _OXA-23_ ^1^		*sul2*	*strA*	*msr(E)*		
*bla*_NDM-1_ ^1^			*strB*			
*bla* _PER-7_			*aph (3′)-VI*			
*bla* _ADC-57_			*ant (3″)-IIa*			

^1^ Recognised carbapenemase genes.

**Table 3 ijms-24-12515-t003:** Virulome of the *A. baumannii* isolate Ale25. Genes are represented in italics. BCRR—Biofilm-controlling response regulator; BAP—Biofilm-associated protein; OMP—Outer-membrane proteins; PNAG—Poly-N-acetyl-D-glucosamine; PBPG—Penicillin-Binding Protein G; LPS—Lipopolysaccharide; T2SS—Type II secretion system; T6SS—Type VI secretion system.

Adherence and Biofilm Production								Immune Modulation			Effector Delivery Systems		Exotoxins	Iron Uptake
Type IV Pili	BCRR	Efflux Pump	Quorum Sensing	Csu Fimbriae	*BAP*	*OMP*	PNAG	Capsule	PBPG	LPS	T2SS	T6SS	Phospholipases C and D	Acinetobactin
*pilB*	*bfmR*	*adeF*	*abaR*	*csuA*	*bap*	*ompA*	*pgaA,*	*tviB*	*pbpG*	*lpxA*	*gspC*	*tssA*	*plc1,*	*barA*
*pilC*	*bfmS*	*adeG*	*abaI*	*csuB*			*pgaB*	*galE*		*lpxB*	*gspD*	*tssB*	*plc2*	*barB*
*gsp0/pilD*		*adeH*		*csuA/B*			*pgaC*	*galU*		*lpxC*	*gspE1*	*tssC*	*plcD*	*bauA*
*pilF*				*csuC*			*pgaD*	*pgi*		*lpxD*	*gspE2*	*hcp/tssD*		*bauB*
*pilG*				*csuD*						*lpxL*	*gspF*	*tssE*		*basA*
*pilH*				*csuE*						*lpxM*	*gspG*	*tssF*		*basB*
*pilI*										*lpsB*	*gspH*	*tssG*		*basC*
*pilJ*											*gspI*	*clpV/tssH*		*basD* *basG*
*pilM*											*gspJ*	*vgrG/tssI*		*basJ*
*pilN*											*gspK*	*tssK*		*entE*
*pilO*											*gspL*	*tssL*		
*pilP*											*gspN*	*tssM*		
*pilQ*											*gspM*	*tagX*		
*pilR*												*tse2*		
*pilS*														
*pilT*														
*pilU*														
*pilV*														
*pilW*														
*pilX*														
*PilY1*														
*fimT*														
*fimU*														
*fimV*														
*tsaP*														

## Data Availability

The assembled genome is available in NCBI under the accession number JANBZS000000000.

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
