# Peer review of "Co-Existence of blaNDM-1, blaOXA-23, blaOXA-64, blaPER-7 and blaADC-57 in a Clinical Isolate of Acinetobacter baumannii from Alexandria, Egypt"

_ijms, 2023, doi:10.3390/ijms241512515_

Round 1
Reviewer 1 Report
The paper describes an interesting case, the isolation of A. baumannii strain carrying five beta-lactamases simultaneously. In general, the work makes a good impression, but some points require clarification and more details.
Comments
· Authors used PCR-based replicon typing (AB-PBRT) to determine the presence of replicase genes, however in 2022 Lam et al proposed a tool for typing and the identification of plasmids in draft genomes of A. baumannii. The authors are familiar with this work because they cite it. It is advisable to analyze genome of Ale25 isolate using the proposed approach.
· The discrepancy between phenotypic susceptibility to gentamicin and tobramycin and the presence of genes determining resistance to these antibiotics needs to be discussed.
· Figure 2 captions as well as conditions and results of electrophoresis need to be clarified.
o If I understand correctly, line 3 in figure 2A corresponds to the Ale25 isolate. It is necessary to mark those bands that the authors consider as plasmids and indicate their molecular weights.
o Characterization of E. coli 678 and 679 isolates is not provided. The possibility of their use as molecular weight markers is not obvious.
o Fig 2C – Method of DNA isolation and electrophoresis is not described in the text, the chromosomal localization of the blaper-7 gene is not obvious, since the hybridization band is at the 21 kb level.
Author Response
REVIEWER 1
Request: Authors used PCR-based replicon typing (AB-PBRT) to determine the presence of replicase genes, however in 2022 Lam et al proposed a tool for typing and the identification of plasmids in draft genomes of A. baumannii. The authors are familiar with this work because they cite it. It is advisable to analyze genome of Ale25 isolate using the proposed approach.
Comment As requested, we have performed the analysis following Lam et al. scheme using the draft genome and the corresponding results have been added to the manuscript.
Request: The discrepancy between phenotypic susceptibility to gentamicin and tobramycin and the presence of genes determining resistance to these antibiotics needs to be discussed.
Comment We performed broth microdilution experiments to ensure these results and the MIC values of the VITEK2 for gentamicin and tobramycin were wrong, Ale25 was resistant to gentamicin and tobramycin. We changed the values for the correct ones.
Request: Figure 2 captions as well as conditions and results of electrophoresis need to be clarified.
If I understand correctly, line 3 in figure 2A corresponds to the Ale25 isolate. It is necessary to mark those bands that the authors consider as plasmids and indicate their molecular weights.
Characterization of E. coli 678 and 679 isolates is not provided. The possibility of their use as molecular weight markers is not obvious.
Fig 2C – Method of DNA isolation and electrophoresis is not described in the text, the chromosomal localization of the blaper-7 gene is not obvious, since the hybridization band is at the 21 kb level.a
Comment To improve conventional gel analysis and have more accurate plasmid results we performed Pulsed Field Gel Electrophoresis and ask for professional bioinformatic analysis of the sequences. That’s why we can confirm that all the Beta-lactamase genes are located in the chromosome except to the blaPER-7 gene, which is located in a plasmid of 240 Kb, one of the four plasmids we found by plasmid sequencing experiments. As a result, we changed Figure 2 (deleting the blaPER-7 gene hybridization) and include a Figure 3 with the complete sequence of the 240 Kb plasmid containing the blaPER-7 gene.
References of E. coli 678 (CECT 678/ NCTC 50193) and 679 (CECT679/NCTC 50192) isolates are provided and included in the footnote of Fig 2A. They are control strains for plasmid analysis and belong to the Spanish Type Culture Collection https://www.uv.es/uvweb/spanish-type-culture-collection/en/cect/strains/culture-media-catalogue-/strains-search-engine-1285892802374.html

Reviewer 2 Report
The authors present the first Egyptian case of A.baumanni isolate, characterised phenotypically as well as genetically, harbouring five β-lactamases . The article is of interest for the scientific community but need some improvements
Major revisions.
1) 1) Line 58/159: identification of the isolate was performed using Vitek 2 system. The system like other automated system are not fully able to correctly identify A.baumanii at a specie level. The reference system is the MALDI TOF assay or by sequencing approach. Please confirm by additional experiments the identification. The antimicrobial susceptibility of the isolates are specie-depending
2) Line 163: The antimicrobial susceptibility testing of the isolate was performed using VITEK 2. While Antimicrobial activity of cefiderocol was determined by disk diffusion method. Both methods are not the reference ones to perform AST. Use the righ method to test AST of the isolate is paraumount, particularly for drugs such cefiderocol and colistin (see EUCAST 2023 Breakpoint tables for interpretation of MICs and zone diameters Version 13.0, valid from 2023-01-01
3) Methods section: what is the standard use to interpretate the AST: Eucast or CLSI? Please add this information
4) the approval of the ethics committee is missing due to the fact that a case referable to a patient is cited
Minor revision:
line 48: please add a reference
Author Response
REVIEWER 2
Request: Line 58/159: identification of the isolate was performed using Vitek 2 system. The system like other automated system are not fully able to correctly identify A.baumanii at a specie level. The reference system is the MALDI TOF assay or by sequencing approach. Please confirm by additional experiments the identification. The antimicrobial susceptibility of the isolates are specie-depending
Comment: As we state in the text, we identified the species by four different methodologies: MALDI-TOF and VITEK; gyrB multiplex PCR; PCR detection of the blaOXA-51 gene; and Whole Genome Sequencing to ensure the identification at a specie level.
Request: Line 163: The antimicrobial susceptibility testing of the isolate was performed using VITEK 2. While Antimicrobial activity of cefiderocol was determined by disk diffusion method. Both methods are not the reference ones to perform AST. Use the righ method to test AST of the isolate is paraumount, particularly for drugs such cefiderocol and colistin (see EUCAST 2023 Breakpoint tables for interpretation of MICs and zone diameters Version 13.0, valid from 2023-01-01
Comment: Following your suggestion, we performed broth microdilution method following CLSI guidelines and using EUCAST clinical breakpoints. Results are included in the text.
Request: Methods section: what is the standard use to interpretate the AST: Eucast or CLSI? Please add this information
Comment: Done.
Request: the approval of the ethics committee is missing due to the fact that a case referable to a patient is cited
Comment: We included the Institutional Review Board Statement and submitted the approval of the ethics committee as requested. We send the original approval documents from the University of the Basque Country and AAST&MT.
Request: line 48: please add a reference
Comment: Done

Round 2
Reviewer 2 Report
The manuscript has been improved. However, there are still some critical issues regarding the interpretation of the antimicrobial susceptibility of isolates. According to EUCAST ticarcillin, Aztreonam, ceftazidime, cefepime should not be tested (-); Ticar/cla, pip, minocycline, tigecycline and cefiderocol are IE (therefore it cannot be said that they are susceptible or resistant). I also believe that it would be better to report the AST (with correct interpretation) in a table
Author Response
REVIEWER 2
Request: The manuscript has been improved. However, there are still some critical issues regarding the interpretation of the antimicrobial susceptibility of isolates. According to EUCAST ticarcillin, Aztreonam, ceftazidime, cefepime should not be tested (-); Ticar/cla, pip, minocycline, tigecycline and cefiderocol are IE (therefore it cannot be said that they are susceptible or resistant). I also believe that it would be better to report the AST (with correct interpretation) in a table
Comment: Done. We have adjusted the interpretation and included a table to report the AST.